# Occurrence of *Clostridium perfringens* in Wild Mammals in the Amazon Biome

**DOI:** 10.3390/ani14091333

**Published:** 2024-04-29

**Authors:** Hanna Gabriela da Silva Oliveira, Ananda Iara de Jesus Sousa, Isabela Paduá Zanon, Cinthia Távora de Albuquerque Lopes, Rodrigo Otavio Silveira Silva, Sheyla Farhayldes Souza Domingues, Felipe Masiero Salvarani

**Affiliations:** 1Instituto de Medicina Veterinária, Universidade Federal do Pará, Castanhal 68740-970, PA, Brazil; hnnagabriela@gmail.com (H.G.d.S.O.); anandaiara@hotmail.com (A.I.d.J.S.); cinthia@ufpa.br (C.T.d.A.L.); shfarha@ufpa.br (S.F.S.D.); 2Laboratório de Bacterioses e Pesquisa da Escola de Veterinária, Universidade Federal de Minas Gerais, Belo Horizonte 31270-901, MG, Brazil; pasuaisabela@gmail.com (I.P.Z.); rodrigo.otaviosilva@gmail.com (R.O.S.S.)

**Keywords:** *Clostridium perfringens*, alpha toxin, toxinotype A, wild mammals, diarrhoea

## Abstract

**Simple Summary:**

*Clostridium perfringens* is a commensal bacterium of humans and other animals that can become pathogenic, causing myonecrosis and enteric diseases. Previous studies have reported *C. perfringens* infection in several wild animals as well as its presence as a commensal, but its role in animals of the Amazon region is largely unknown. Thus, this study investigated the occurrence of *C. perfringens* in samples from wild mammals treated at the Wild Animals Sector of the Veterinary Hospital of the Federal University of Pará in the Amazon biome. The results demonstrate that *C. perfringens* type A was the only toxinotype isolated from mammals in the Amazon biome, demonstrating the importance of further studies on the topic in wild animals, their impacts and the diseases caused.

**Abstract:**

The objective of this study was to evaluate the occurrence of *Clostridium perfringens* in stool samples and swabs collected from wild mammals in the Amazon biome. Sixty-five faecal and swab samples were collected in situ and ex situ from 16 species and three genera of wild mammals, some of which were in good health and some of which had diarrhoea. After pre-enrichment, the samples were plated on selective agar for *C. perfringens*. Characteristic colonies were subjected to *multiplex* PCR for the detection of genes encoding the main *C. perfringens* toxins (alpha, beta, epsilon, and iota toxin and enterotoxin). Among the 65 samples, 40 (61.5%) were positive for the gene encoding the alpha toxin and were classified as type A, 36 of which were asymptomatic animals and four were diarrheal. No other toxinotypes were found. The findings of this study suggest that *C. perfringens* type A is commonly found in mammal species of the Amazon biome. This seems to be the first study to identify *C. perfringens* type A in species such as *B. variegatus* (common ground sloth), *C. didactylus* (two-toed sloth), *P. flavus* (Jupará), *T. tetradactyla* (anteater), *S. collinsi* (squirrel monkey), *S. niger* (black marmoset), and *S. apella* (Guyana capuchin) and in the genus *Didelphis* sp. (opossum).

## 1. Introduction

*Clostridium perfringens* is a Gram-positive, spore-forming bacillus that is classified into seven toxinotypes (A–G) according to the production of six major toxins: alpha (*cpa*), beta (*cpb*), epsilon (*etx*), iota toxin (*itx*), enterotoxin (*cpe*) and necrotic enteritis type B (*netB*) [1,2,3]. *C. perfringens* is found mainly as a commensal of the gastrointestinal tract of humans and animals and is ubiquitous in the environment [3].

Despite being a common commensal, *C. perfringens* can cause a number of diseases in animals and humans, including gas gangrene and various enteric conditions [1,4]. In humans, it is the second most common cause of foodborne illness in the United States and the sixth most common in Brazil [5]. Some studies have demonstrated the presence of *C. perfringens* as a commensal in some species of wild animal in addition to confirming the participation of this agent as a cause of enteritis and enterotoxaemia [6,7,8]. However, the description of this bacterium and its toxinotypes in wild animals of the Amazon biome, either as a commensal or as a pathogen, is lacking.

It remains largely unknown which species are colonized by this agent, and among those that have *C. perfringens* as a commensal species, it is not known which toxinotypes are present or which toxins those strains can produce. Thus, the objective of this study was to evaluate the isolation frequency and types of *C. perfringens* in wild animals in the Amazon biome.

## 2. Materials and Methods

### 2.1. Data Collection

This study was conducted in accordance with the Ethical Principles of Animal Research adopted by the National Council for the Control of Animal Experimentation (CONCEA) and was approved by the Committee on Ethics in the Use of Animals of the Federal University of Pará (CEUA/UFPA) under protocol n. 8888280618. The study was also authorized by the Chico Mendes Institute for Biodiversity Conservation (ICMBio) under protocol n. 67300-1.

A non-probabilistic sampling method (for convenience) was used [9] due to all the particular characteristics of the species of the animals treated and hospitalized during the period from August 2017 to October 2022 at the Veterinary Hospital (Wild Animals Sector) of the Federal University of Pará, located in the city of Castanhal, state of Pará. The collection procedures (Appendix A) were monitored by veterinarian residents to monitor animal welfare, including the physical, mental and behavioural states of each species [10].

Stool samples were collected during feeding, which occurred twice a day (morning and afternoon), to reduce contact and handling stress. Pieces of PVC film paper were placed under the animals’ enclosure, with subsequent collection of fresh, individual faeces produced without much soiling of the substrate. The collection of samples were also collected during medication applications, physical examinations and weighing [11]. Cloacal and anal swabs (Appendix A) were collected under the same conditions, and proper physical and chemical protocols for each species were applied to reduce the risk of accidents and stress [12].

The stool samples were placed in microtubes, and the swab samples were transferred to a saline solution (Appendix A), then labelled with the animal identification, species and collection date. After labelling, the samples were stored at −20 °C [13].

### 2.2. Processing

All samples were sent to the Laboratory of Bacteriosis and Research of the Federal University of Minas Gerais (UFMG) for research and typing of *C. perfringens* [13]. For the isolation of *C. perfringens*, between 0.08 and 0.12 g of faeces were immersed in 1 mL of brain heart infusion broth (BHI, Difco Laboratories, Detroit, MI, USA) for enrichment and incubated in an anaerobic chamber (Thermo Scientific, Waltham, MA, USA) for 24 h at 37 °C. Cloacal and anal swabs were inoculated directly into tubes with BHI and subjected to the same culture conditions. After incubation, 10 μL aliquots of each sample were plated on Shahadi–Ferguson perfringens agar (Thermo Fisher, Waltham, MA, USA) and incubated again in an anaerobic chamber at 37 °C for 48 h [13].

For each plate, up to three characteristic colonies containing sulphite-reducing microorganisms (Appendix A) were suspended in 400 μL of ultrapure water (Milli-q^®^ iq 7000 purification system, Sigma-Aldrich, Burlington, MA, USA in Eppendorf tubes (2 mL) and subjected to thermal DNA extraction at 98 °C for 20 min in a thermoblock (Sigma-Aldrich, Burlington, MA, USA) [13,14]. Subsequently, the microtubes were centrifuged at 3000× *g* for 10 min, and the resulting supernatant was used as template DNA in a multiplex PCR to identify the genes encoding the alpha, beta, epsilon and iota toxins and enterotoxin. The primers used are in Table 1. In the multiplex PCR, 35 cycles were used, with an initial cycle of denaturation at 94 °C for five minutes, followed by 34 cycles at 94 °C for one minute for denaturation, 48 °C for one minute for annealing and 72 °C for one minute of extension. A final extension cycle at 72 °C for seven minutes was also included. A final volume of 50 μL was used, using 1.5 mM MgCl_2_, 25 pmoles of each primer, 0.2 mM dNTPS and 2.5 U of Taq polymerase. For all the PCRs, the amplifications were performed in a thermocycler (Thermal Cycler Px2, Thermo Fisher, Waltham, MA, USA), and the bands were visualized with ultraviolet light on a 2% agarose gel stained with ethidium bromide (Sigma-Aldrich, Burlington, MA, USA) [13,14]. Positive and negative controls were used in the test to validate the experiment.

### 2.3. Statistical Analyses

The data were summarized using frequency tables. To measure the association between the categorical variables (presence of diarrhoea, host, sample type [swabs or faeces]) and isolation of *C. perfringens*, a univariate analysis was performed using Fisher’s exact test with a significance level of *p* ≤ 0.05. The odds ratios with 95% confidence intervals were also calculated. All analyses were performed using R Software 4.0.9 (R Foundation for Statistical Computing, Vienna, AUT.).

## 3. Results

Sixty-five stool samples and cloacal or anal swabs were obtained from 16 species and three genera of mammals in good health or with diarrhoea (Table 2). Among the species included in the study, five were the most common: *Nasua nasua* (South American coati), with 14 samples (21.4%); *Bradypus variegatus* (common ground sloth), with eight samples (12.3%); *Didelphis* sp. (opossum), with seven samples (10.8%); and *Choloepus didactylus* (sloth), *Saimiri collinsi* (squirrel monkey), and *Sapajus apella* (Guyana capuchin), each with five samples (7.7%). Fifty nine (90.8%) of the samples were from healthy individuals, and six (9.2%) were from individuals with diarrhoea. The stool samples (32/65 = 49.2%) were balanced with the anal and cloacal swabs (33/65 = 50.8%). Fewer animals (19, 29.2%) were sampled in situ than the number sampled ex situ (46, 70.8%). The multiplex PCR used to identify the *C. perfringens* toxinotypes in the study is illustrated in Appendix A.

*C. perfringens* was isolated from 40 (61.5%) samples, 36 (90.0%) samples from asymptomatic animals and four (10%) from sick animals with clinical signs of diarrhoea. All the isolates were positive only for the gene encoding the alpha toxin and were classified as type A (Table 2). Approximately half of the healthy animals (36/59 = 55.4%) were positive for *C. perfringens* type A, while four of the six animals with diarrhoea (4/6 = 66.7%) were positive for this toxinotype, with no difference between groups (*p* = 1.0). The animals that were positive for *C. perfringens* type A and that presented diarrhoea were of the genus *Didelphis* sp. (opossum) and of the species *Sapajus apella* (Guyana capuchin), *Puma yagouaroundi* (buckwheat) and *Nasua nasua* (South American coati).

The isolation of *C. perfringens* was greater in the stool samples than in the swab samples (*p* = 0.0105): a stool sample was approximately four times more likely to be positive than a swab sample (OR = 4.28; CI: 1.47–12.6). There was no difference in the frequency of isolation between ex situ and in situ animals (*p* = 0.16), with thirty one (47.7%) and nine (13.8%) positive animals, respectively. The difference between 47.7% and 13.8% is large, so the reason the difference is not statistically significant is very likely due to the small sample size.

Samples from *Alouatta caraya* (black howler monkey, n = 1), *Saguinus ursulus* (marmoset, n = 1), *Mazama americana* (Mateiro deer, n = 1), *Mazama nemorivaga* (brown deer, n = 1) and *Dasypus novemcinctus* (nine-banded armadillo, n = 1) were negative for *C. perfringens* in this study.

## 4. Discussion

Most studies investigating the prevalence of *C. perfringens* in wild animals focused on a single species or on species from the same family and with similar diets [13,15,16]. This appears to be the first study covering this diversity of forest mammals within Brazil, allowing a better understanding of the occurrence of *C. perfringens* in the Amazon biome.

The finding that more than 60% of the samples were positive for *C. perfringens* type A, thirty six (90.0%) samples from asymptomatic animals and four (10%) from sick animals with clinical signs of diarrhoea corroborates the results in other wild animal species, which show that type A is the most prevalent commensal [13,15,16,17]. The absence of this anaerobic microorganism in some species might be due to the low sampling rate of some species, especially for n = 1. Another study also hypothesised the non-colonization or low colonization of some species.

In addition, in this study, cloacal swabs had a lower rate of *C. perfringens* isolation than faeces and excreta, which could have influenced *C. perfringens* detection in some species. Interestingly, there was no difference between the isolation rate of *C. perfringens* from faeces and that from cloacal swabs in a study of toucans in Brazil. It is widely accepted in medical and veterinary practice that faecal samples are more reliable than rectal swabs for identifying gut bacteria. The lack of difference in toucans could be the use of cloacal rather than rectal swabs. Faeces are stored in the cloaca, while in many eutherian mammals, the faeces can pass quickly through the lower rectum rather than being stored there. Furthermore, it is worth noting that most of the stool samples obtained in the present study were collected ex situ, representing more than two-thirds of the samples obtained. This pattern could have affected the rate of isolation, since studies have suggested that *C. perfringens* is apparently more likely to be observed in ex situ samples than in in situ samples [15].

Despite the importance of *C. perfringens* as an enteropathogen in domestic animals and humans, the actual role of type A in diarrhoeal diseases remains uncertain in several wild species due to the absence of a marker that allows for the differentiation of *C. perfringens* when it is present as an enteropathogen or as a commensal [13,15]. Nevertheless, there are reports suggesting that type A is the cause of lethal haemorrhagic enteritis in several species, such as *Panthera tigris altaica* (Siberian tiger), *Panthera leo* (lion) [8], *Vulpes vulpes* (red fox) [18], *Loxodonta africana* (elephant) [19], *Selenarctos thibetanus* (Asian black bear) [20] and *Papio hamadryas* (hamadryas baboon) [21]. Previous studies have suggested that factors other than the toxinotype may be associated with the occurrence of enteric disease caused by *C. perfringens* type A in animals and humans, such as stress, the dysregulation of the intestinal microbiota, and parasitic and viral infections [5,15,18,20,21,22,23,24,25,26,27,28]. In other words, *C. perfringens* type A is a facultative bacterium, meaning it is primarily commensal but can cause disease if the gut environment is disturbed or the host is stressed or otherwise immunocompromised.

The results of the present study corroborate previous findings suggesting that *C. perfringens* type A is a commensal in *N. nasua* (coati) [13], *C. thous* (crab-eating fox), *P. yagouaroundi* (black-tailed cat), and *L. pardalis* (ocelot) [16]. On the other hand, this seems to be the first report of *C. perfringens* in some species, such as *B. variegatus* (common ground sloth), *C. didactylus* (two-toed sloth), *T. tetradactyla* (anteater), *Didelphis* sp. (opossum), *S. apella* (Guyana capuchin), *S. collinsi* (squirrel monkey), *S. niger* and *P. flavus* (Jupará).

Studying the presence and impact of *Clostridium* and other pathogens in wild animals from the Amazon biome is crucial for understanding the pathogens present in wild animal populations and helps to assess the overall health of the ecosystem. Changes in pathogen prevalence or emergence of new pathogens can indicate disruptions in the ecosystem. In the conservation of the biodiversity of animal life, many wild animal species in the Amazon are already facing threats such as habitat loss, climate change and poaching. Diseases caused by pathogens like Clostridium can further endanger these populations. Studying these pathogens can help develop conservation strategies to protect vulnerable species. Studying pathogens in wild animals is in line with the One Health approach, which recognizes the interconnectedness of human, animal and environmental health. By studying pathogens in wild animals, researchers can gain insights into how diseases spread across different species and ecosystems. Another important point in future research is antimicrobial resistance, since Clostridium and other pathogens can develop antimicrobial resistance, posing a threat to both animal and human health. Studying these pathogens in wild animals can help monitor and mitigate the spread of antimicrobial resistance. However, it is necessary to be very clear about all the difficulties and challenges for the future of research with wild animals. To conduct these studies effectively, researchers would need to collaborate across disciplines, including veterinary medicine, wildlife biology, ecology and public health. Also, a very important factor is that researchers would also need to consider the challenges of working in remote and diverse environments like the Amazon biome, such as limited access to samples and logistical challenges. Nonetheless, the insights gained from such studies would be invaluable for the health of both wild animals and humans.

## 5. Conclusions

Given the rich biodiversity of the Amazon biome and the diverse range of mammalian species that inhabit it, it is likely that *C. perfringens* is present in many of these animals. The findings of the present study demonstrated that *C. perfringens* type A was the only toxinotype found in the sampled animals, being mostly isolated from apparently healthy animals, and can be considered as commensal. However, under certain conditions, such as when the animal is stressed or if there are changes in diet, the bacteria can multiply rapidly and produce toxins that can cause disease. Therefore, more research is needed to fully understand the prevalence and impact of this bacterium on wild mammal populations in the Amazon.

## Figures and Tables

**Table 1 animals-14-01333-t001:** Specific primers used to amplify the alpha, beta, epsilon, iota and enterotoxin genes of *C. perfringens* toxinotypes.

Genes	Primers	Fragment Size
Alpha (*Cpa*)	5′GCTAATGTTACTGCCGTTGA 3′sense) 5′CCTCTGATACATCGTGTAAG 3′(anti-sense)	324 bp
Beta (*Cpb*)	5′GCGAATATGCTGAATCATCTA 3″(sense) 5′GCAGGAACATTAGTATATCTTC 3′(anti-sense)	196 bp
Epsilon (*Etx*)	5′GCGGTGATATCCATCTATTC 3′(sense) 5′CCACTTACTTGTCCTACTAAC 3′(anti-sense)	655 bp
Iota (*iA*)	5′TTTTAACTAGTTCATTTCCTAGTTA 3′(sense) 5′TTTTTGTATTCTTTTTCTCTAGATT 3′(anti-sense)	298 bp
Enterotoxin (*Cpe*)	5′GGAGATGGTTGGATATTAGG 3′(sense) 5′GGACCAGCAGTTGTAGATA 3′(anti-sense)	233 bp

**Table 2 animals-14-01333-t002:** Number of samples (%) of the species and genera of mammals collected and toxinotype identified.

Species	Common Name	Samples (%)	Faeces	Swabs	Asymptomatic Patient	Diarrhoeal Patient	Ex Situ	In Situ	Toxinotype
*B. variegatus*	Common ground sloth	8 (12.3%)	3 (3)	5 (3)	8 (6)	0	5 (4)	3 (2)	A
*C. didactylus*	Two-toed sloth	5 (7.7%)	3 (3)	2 (0)	5 (3)	0	4 (3)	1 (0)	A
*P. flavus*	Jupará	1 (1.5%)	0	1 (1)	1 (1)	0	1 (1)	0	A
*S. collinsi*	Squirrel monkey	5 (7.7%)	1 (1)	4 (2)	4 (3)	1 (0)	3 (2)	2 (1)	A
*S. niger*	Black marmoset	1 (1.5%)	1 (1)	0	1 (1)	0	1 (1)	0	A
*S. appeal*	Guiana capuchin monkey	5 (7.7%)	4 (3)	1 (0)	4 (2)	1 (1)	2 (2)	3 (1)	A
*A. caraya*	Howler monkey	1 (1.5%)	1 (0)	0	1 (0)	0	1 (0)	0	-
*S. ursulus*	Marmoset	1 (1.5%)	0	1 (0)	1 (0)	0	1 (0)	0	-
*C. thous*	Crab-eating fox	2 (3.1%)	1 (1)	1 (0)	2 (1)	0	2 (1)	0	A
*P. yagouaroundi*	Black-tailed cat	3 (4.6%)	2 (2)	1 (1)	2 (2)	1 (1)	2 (2)	1 (1)	A
*L. pardalis*	Ocelot	3 (4.6%)	1 (1)	2 (0)	3 (1)	0	2 (1)	1 (0)	A
*T. tetradactyla*	Lesser anteater	3 (4.6%)	2 (1)	1 (1)	3 (2)	0	3 (2)	0	A
*N. nasua*	South American coati	14 (21.5%)	6 (4)	8 (4)	12 (7)	2 (1)	8 (5)	6 (3)	A
*A. Mazana*	Bush deer	1 (1.5%)	1 (0)	0	1 (0)	0	1 (0)	0	-
*M. nemorivaga*	Brown brocket	1 (1.5%)	1 (0)	0	1(0)	0	1 (0)	0	-
*D. novemcinctus*	Nine-banded armadillo	1 (1.5%)	0	1 (0)	1 (0)	0	1 (0)	0	-
*Sapajus* sp.	Capuchin monkey	1 (1.5%)	0	1 (1)	1 (1)	0	1 (1)	0	A
*Saguinus* sp.	Marmoset	2 (3.1%)	0	2 (1)	2 (1)	0	2 (1)	0	A
*Didelphis* sp.	Opossum	7 (10.8%)	5 (5)	2 (1)	6 (5)	1 (1)	5 (5)	2 (1)	A
	Total	65	32 (25)	33 (15)	59 (36)	6 (4)	46 (31)	19 (9)	

## Data Availability

Data are contained within the article.

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
