# Peer review of "Occurrence of Clostridium perfringens in Wild Mammals in the Amazon Biome"

_animals, 2024, doi:10.3390/ani14091333_

Round 1

Reviewer 1 Report

Comments and Suggestions for Authors

This is a very nice short study which covers the isolation and characterisation of toxins from C. perfringens from wild animals in the Amazon. It is nicely and very succinctly written, and offers some new and important information on pathogen carriage by multiple species-although numbers are small- and some interesting information on the best samples to take.

I have only a few minor comments which I have detailed below.

Line 29- can delete one of the ‘more’ words in here to make the sentence flow better

Keywords- maybe worth including C. perfringens in here?

Line 39- Gram needs to be capitalised

Line 94- was a PCR carried out to confirm that the isolates were C. perfringens?

Line 94- probably worth including the PCR reagents in here too- unless identical to the referenced work.

Table 1 is tricky to understand as a number of the abbreviations are not included in the legend. Please can you include them all?

Line 131- Isolation doesn’t need a capital letter

Author Response

Dear Reviewer 1,
We would like to thank you for reviewing our article, suggesting important corrections to improve the article and, furthermore, for recognizing, in your words that "This is a very nice short study which covers the isolation and characterization of toxins from C. perfringens from wild animals in the Amazon. It is nicely and very succinctly written, and offers some new and important information on pathogen carriage by multiple species-although numbers are small- and some interesting information on the best samples to take".

Below, point by point, I report your suggestions:
1). Line 29- can delete one of the ‘more’ words in here to make the sentence flow better. Correction made.
2). Keywords- maybe worth including C. perfringens in here? Correction made.
3). Line 39- Gram needs to be capitalized. Correction made. Sorry for this error.
4). Line 94- was a PCR carried out to confirm that the isolates were C. perfringens? Yes. All samples were conformed via multiplex PCR as shown in lines 95 and 96 "as template DNA in a multiplex PCR to identify the genes encoding the alpha, beta, epsilon and iota toxins and enterotoxin [16]."
5). Line 94- probably worth including the PCR reagents in here too- unless identical to the referenced work. Correction made.
6). Table 1 is tricky to understand as a number of the abbreviations are not included in the legend. Please can you include them all? Correction made. Sorry for this error.
7). Line 131- Isolation doesn’t need a capital letter. Correction made.

Thank you very much!
Sincerely,

Felipe Masiero Salvarani

Reviewer 2 Report

Comments and Suggestions for Authors

Please refer to the attached file for detailed comments and suggestions.

Author Response

Dear Reviewer 2,
We would like to thank you for your willingness to review our manuscript and for recognizing that the article brings the scientific community reporting of C. perfringens type A from several mammals for the first time.

Below I will detail your questions and suggestions point by point:

2) General concept comments an important weakness is that Table 1. is missing important details (see below). Another weakness is the low number of animals in many species (13 species have fewer than 5 samples, and 8 species have only 1 sample each). Although 6 animals showed diarrhea, there is no more information provided about these animals that might indicate the severity of the disease and any other factors that might have contributed to the disease. So
the reader cannot judge the significance of the presence of C. perfringens in these animals. Regarding table 1, we tried to make it as comprehensive and didactic as possible, it brings all the information we had available at the Veterinary Hospital, wild animal sector, at the Federal University of Pará. I am sorry that there is no way to improve it, but we believe that this is not be a weak point and I reiterate that our opinion was also shared by the other three reviewers, who, like you, reviewed the article. Now highlighting the other point raised about the low number of animals in many species (13 species have fewer than 5 samples, and 8 species have only 1 sample each), it must be clear that working with wild animals is an extremely challenging area, because despite being in the Amazon Biome, the routine reception of wild animals is still low, in addition, in the vast majority of cases these wild animals come from seizures by police authorities due to the trafficking of wild animals. And when these animals arrive to us at the Veterinary Hospital, where I am one of the veterinarians in the wild sector, they are very weak and therefore our main focus is to try to stabilize the animal and keep it alive, collecting biological samples. , like feces, as a background factor. Many of the animals do not survive and those that survive after the first care provided by us are kept in "UTAs" (intensive treatment units) and we try to manipulate these animals as little as possible, as it is public knowledge that stress is one of the main factors that increase the chances of death of wild animals. Therefore, we collect biological materials such as feces, only when the animal is no longer at risk of death. And the quantity by species ends up varying depending on what I have just explained, as I can only collect the animal as it will be sent for release or even die. For others who spend more time in the Veterinary Hospital, we have the opportunity to carry out more collections. In addition, some species occur or are more common in trafficking seizures than others. I hope I was able to explain.

3). Line 14: Simple Summary: “Clostridium perfringens is a commensal bacterium of humans and animals that can act as a pathogen that causes myonecrosis and enteric diseases.” Suggest change to: “Clostridium perfringens is a commensal bacterium of humans and other animals that can become pathogenic, causing myonecrosis and enteric diseases.” A scientific paper should identify humans as an animal species. Correction made.

4). Line 53: “it is not known which toxinotypes are present or which toxins are potentially produced by those strains. Thus, the objective of the present study was” Suggest change to: “it is not known which toxinotypes are present or which toxins those strains can produce. Thus, the objective of this study was”. More succinct English. Correction made.

5). Line 73: “The collection of samples also occurred during medication applications “ Suggest change to: “Samples were also collected during medication applications “. More succinct English. Correction made.

6). Line 76: “were applied to reduce the risk of accidents and stress.”Suggest change to: “were applied to reduce the risk of accidents and stress.” Typographical error. Correction made.

7). Line 78: “to a saline solution, after which the samples were labeled with the animal identification, species and collection date. After labeling, the samples were stored in a refrigerator at -20 °C [15].” Suggest change to: “to a saline solution, then labeled with the animal identification, species and collection date. After labeling, the samples were stored at -20 °C [15].” More succinct English. Correction made.

8). Line 114: “In the present study, 59 (90.8%) of the samples were from healthy individuals, and six (9.2%) were from individuals with diarrhea. Regarding the type of sample, there was a balance between stool samples (32/65=49.2%) and anal and cloacal swabs (33/65=50.8%). The number of animals sampled in situ (19, 29.2%) was lower than the number of animals sampled ex situ (46, 70.8%).”
Suggest change to: “Fifty nine (90.8%) of the samples were from healthy individuals, and six (9.2%) were from individuals with diarrhea. The stool samples (32/65=49.2%) were balanced with the anal and cloacal swabs (33/65=50.8%). Fewer animals (19, 29.2%) were sampled in situ than the number sampled ex situ (46, 70.8%).”More succinct English. Correction made.

9). Line 120: Table 1. What do “AND (*)” and “Y” designate? They are not explained in the text or Legend. Although S (swab) and E (ex situ) are included in the Legend they do not appear in the Table. They should be included. Correction made.

10). Line 133: “There was no difference in the frequency of isolation between ex situ and in situ animals (p=0.16), with 31 (47.7%) and nine (13.8%) positive animals, respectively.” The difference between 48% and 14% is large so the reason the difference is not statistically significant is very likely due to the small sample size. This should be addressed. Correction made.

11). Line 138: “were negative for C. perfringens in the present study.”
Suggest change to: “were negative for C. perfringens in this study.” More succinct English. Correction made.

12). Line 141: “Most previous studies that investigated the prevalence of C. perfringens in wild animals have focused on a single animal species or on groups belonging to the same family and with similar feeding habits.” Suggest change to: “Most studies investigating the prevalence of C. perfringens in wild animals focused on a single species or on species from the same family and with similar diets.” More succinct English. Correction made.

13). Line 147: “corroborates the findings of previous studies of other species of wild animals, showing that type A is the most prevalent commensal.” Suggest change to: “corroborates the findings in other wild animal species, which show that type A is the most prevalent commensal.” More succinct English. Correction made.

14). Line 148: “Some species included in the present study were negative for this anaerobic microorganism. This result may be related to the low sampling rate of some animals, especially for those with only one sample. A previous study also proposed the hypothesis of non-colonization or low colonization of some species.” Suggest change to: “The absence of this anaerobic microorganism in some species might be due to the low sampling rate of some species, especially for n = 1. Another study also hypothesized non-colonization or low colonization of some species.” More succinct English. Correction made.

15). Line 153: “In addition, the present study revealed that swabs had a lower rate of C. perfringens isolation than faeces and excreta, which may also have influenced C. perfringens detection in some species. Interestingly, a study conducted with toucans in Brazil revealed no difference between the isolation rate of C. perfringens from faeces and that from swabs. Suggest change to: “In addition, in this study cloacal swabs had a lower rate of C. perfringens isolation than faeces and excreta, which could have influenced C. perfringens detection in some species. Interestingly, there was no difference between the isolation rate of C. perfringens from faeces and that from cloacal swabs in a study of toucans in Brazil.” Correction made.

16). It is widely accepted in medical and veterinary practice that faecal samples are more reliable than rectal swabs for identifying gut bacteria. The lack of difference in toucans could be the use of cloacal rather than rectal swabs. Faeces are stored in the cloaca while in many eutherian mammals the faeces can pass quickly through the lower rectum rather than being stored there. Unfortunately, Table 1 does not show any results for swabs (S) despite the inclusion of “S” in the Legend, so it is not clear whether the samples from the opossum were from faeces or the cloaca and so whether they were more like birds or the other mammals in reliability. Does “Y” designate cloacal swabs? If the authors wished to compare the reliability of faecal samples with rectal or cloacal swabs they should have taken swabs from all animals that they could. Then they might have been able to compare the reliability of the two methods. Sorry for our mistake, they really should have taken swabs from all animals that they could, but unfortunately it wasn't done. And if this reviewer doesn't mind, we'll insert his precise and important comment "It is widely accepted in medical and veterinary practice that faecal samples are more reliable than rectal swabs for identifying gut bacteria. The lack of difference in toucans could be the use of cloacal rather than rectal swabs. Faeces are stored in the cloaca while in many eutherian mammals the faeces can pass quickly through the lower rectum rather than being stored there."

17). Line 173: “Previous studies have suggested that factors other than the toxinotype may be associated with the occurrence of enteric disease caused by C. perfringens type A in animals and humans, such as stress, dysregulation of the intestinal microbiota, and parasitic and viral infections .” I was taught that C. perfringens type A is a facultative bacterium, meaning it is primarily commensal but can cause disease if the gut environment is disturbed or the host is stressed or otherwise immunocompromised. This is consistent with these studies and is well known. We agree with your statement and humbly insert the excerpt "In other words, C. perfringens type A is a facultative bacterium, meaning it is primarily commensal but can cause disease if the gut environment is disturbed or the host is stressed or otherwise immunocompromised.", in the discussion as a way of complementing and reaffirming this information.

18). The certificate of approval by the Ethic Committee on Animal Use of the Federal University of Para states that samples were authorized from 20 birds and 20 reptiles. Were they sampled as well? If so, why are they not included in this report? What about the in-situ mammals? No samples from birds and reptiles were collected, as these species were not admitted to the Veterinary Hospital during the project, only mammals. The group of researchers and veterinarians hoped that we would have samples of birds and reptiles, but we did not. That's why the article was based on mammal samples, the only ones collected in the project. Does sampling from them require Ethic Committee approval? The Ethic Committee of which I am a full member authorized the collection of biological samples from wild animals in general, that is, samples from animals in situ or ex situ.

19). Although the study reports the only toxinotype identified was the C. perfringens Type A it is not clear which of the other toxinotypes were tested for although it seems assumed that all types were tested for as it states “No other toxinotypes were found". Correctly, in the rewritten methodology we describe the Multiplex PCR, the primers and the entire technique used to identify the five toxinotypes of C. perfringens. And supplementary figures were inserted in the article, such as, for example, the 1% agorose gel of the Multiplex PCR, which achieves identify the five toxinotypes of C. perfringens at the same time. However, in our samples only toxinotype A was isolated and that is why we mention not having isolated other toxinotypes.

I would like to thank you very much for your patience in creating a supplementary document with all your suggestions, observations and criticisms, as they were fundamental for improving the manuscript.

Sincerely,

Felipe Masiero Salvarani.

Reviewer 3 Report

Comments and Suggestions for Authors

Dear authors thank you for submitting manuscript entitled "Occurrence of Clostridium perfringens in wild mammals in the Amazon biome".

Please address following :

1) Please rewrite the abstract clearly mentioning positive number of samples and percentage from good healthy animal and diarrhoeic animal seperately.

2) Please add primers, PCR conditions and Gel Image in the manuscript.

3) Please show Amazon map and and the place from where you collected sample in the map and add to manuscript.

4) Please add sample collections image from wild mammal or any image or any activities related to collection activity or anything which you think is helpful, add in supplementary file. 

5) Please add image of culture colony plate in supplementary file.

6) Please rearrange the table, make it more simple and easily reflacting of result. or don't abbreviate and write full titile or somehow make it easy to read.

7) have you mention in result as well as in discussion regarding positive percentage from healthy and diseased animal seperately ? Please write it and discussed it like that.

8) Please rewrite conclusion portion.

9) have you try to isolate any other bacteria from diseased animal or any viral pathogen? How can you tell that the diarrhoea is due to solely Clostridium perfringens only, no any other pathogen involved even in negative sample!

10) Since the majorly the isolation of Clostridium perfringens is from the healthy animal, Title of the manuscript need revision.

Comments on the Quality of English Language

Minor

Author Response

Dear Reviewer 2,
We thank you for reviewing our article, suggesting important corrections to improve the article.

Below, point by point, I report your suggestions:
1) Please rewrite the abstract clearly mentioning positive number of samples and percentage from good healthy animal and diarrheal animal separately. Correction made.
2). Please add primers, PCR conditions and Gel Image in the manuscript. Correction made.
3). Please show Amazon map and and the place from where you collected sample in the map and add to manuscript. Dear reviewer, I don't know, it wasn't clear, but when we mention the Amazon Biome, we are not referring to the Amazon, but to all the states that make up this Biome. All samples were collected at the Veterinary Hospital (Wild Animals Sector) of the Federal University of Pará, located in the city of Castanhal, state of Pará. And the Veterinary Hospital receives animals seized by Brazilian inspection bodies, such as IBAMA, Environmental Police, among others. others. That's why we don't have traceability of the samples, we just know that they all come from the Amazon Biome.
4) Please add sample collections image from wild mammal or any image or any activities related to collection activity or anything which you think is helpful. Add in supplementary file.
5) Please add image of culture colony plate in supplementary file. Add in supplementary file.
6). Please rearrange the table, make it more simple and easily reflecting of result. or don't abbreviate and write full title or somehow make it easy to read. Modification made.
7). Have you mentioned in result as well as in discussion regarding positive percentage from healthy and diseased animal separately? Please write it and discuss it like that. Modification made.
8). Please rewrite conclusion portion. Modification made.
9). Have you tried to isolate any other bacteria from diseased animals or any viral pathogen? How can you tell that the diarrhea is due to solely Clostridium perfringens only, no other pathogen involved even in negative sample! Your question is pertinent, but our result does not affirm that diarrhea in the sampled animals is caused by C. perfringens, but rather shows that the majority of C. perfringens isolates were obtained from apparently healthy animals. Our objective was for the first time in the world to show whether C. perfringens occurred in wild mammals in the Amazon Biome. And for this we only work with the isolation of Clostridium perfringes, which is publicly known to be one of the agents considered enteropathogenic, with diarrhea as a clinical sign. In our work, we separated apparently healthy animals, those that did not show any clinical manifestations, from animals with clinical signs, and in this case we chose diarrhea. We really cannot say that other bacteria, viruses or even parasites (Eimeria, Strongyloides, Cryptosporidium) are the causes of the clinical picture of diarrhea, as this differential diagnosis was not made. I reiterate that the division was made in apparently healthy animals, which were the majority and presented C. perfringes type A, probably commensal as in other domestic animals of animals with some clinical signs, in this case diarrhea, as Clostrdium is an enteropathogen.
10). Since most of the isolation of Clostridium perfringens is from the healthy animal, Title of the manuscript needs revision. In the title we did not specify that we would study sick or healthy animals, we limited ourselves to doing what we proposed, which was investigating the occurrence of Clostridium in wild mammals in the Amazon biome. However, we remain open to your suggestion to change the title, because in our opinion the current title does not harm the manuscript and rather reflects the proposed objective.

Thank you very much!
Sincerely,

Felipe Masiero Salvarani 

Reviewer 4 Report

Comments and Suggestions for Authors

da Silva Oliveira et al conducted a study on the occurrence of Clostridium perfringens in wild mammals, which will contribute to the understanding on the distribution of C. perfringens in various animals, especially for wild animals, and thus helps to prevent and control this pathogen. However, there are some problems need to be solved ahead of the acceptance for publication.

Simple summary: The present work only found type A in those animals, but it can not be concluded that "C. perfringens type A is a common commensal bacterium of animals in the Amazon biome", since limited samples were tested, and more work are needed. Revise the sentence to give a more resonable suggestion.

 Summary: 

1. Line 29-30, "Faecal samples were more than four times more likely to be positive for C. perfringens than were swab samples", it's neaningless to compare the two samples, delete it.

2. Repeat sentences were found in both Simple summary and Summary, such as lines 28-30. Revise them.

Keywords: "Clostridium perfringens" should be included as a keyword.

Materials and Methods: did the authors collected samples for one animal for more than one time? or just sampled once?

Results:

1. Table 1: used Three-line table; "S-swab" in the legend was absent in the context of table 1.

2. Only type A was found in this study, which was likely caused by sample resources and no. of samples. Limited samples can not reflect the diversity of types.

Discussion: This part should include the weakness of this study and work needs to be done in the future.

Comments on the Quality of English Language

Moderate editing of English language required

Author Response

Dear Reviewer 4,
We thank you for reviewing our article, suggesting important corrections to improve the article.

Below, point by point, I report your suggestions:
1). Simple summary: The present work only found type A in those animals, but it can not be concluded that "C. perfringens type A is a common commensal bacterium of animals in the Amazon biome", since limited samples were tested, and more work are needed. Review the sentence to give a more reasonable suggestion. Modification made, as well as at the conclusion of the work.
two). Line 29-30, "Faecal samples were more than four times more likely to be positive for C. perfringens than were swab samples", it's impossible to compare the two samples, delete it. Correction made.
3). Repeat sentences were found in both Simple summary and Summary, such as lines 28-30. Review them. Correction made
4). Keywords: "Clostridium perfringens" should be included as a keyword. Correction made.
5). Materials and Methods: did the authors collect samples for one animal for more than one time? or just sampled once? Just sampled once.
6). 1. Table 1: used Three-line table; "S-swab" in the legend was absent in the context of table 1. Table modified, suggestion made by all reviewers.
7). 2. Only type A was found in this study, which was likely caused by sample resources and no. of samples. Limited samples cannot reflect the diversity of types. We agree that the sample size was limited to 60 samples, but other studies with wild and domestic animals also demonstrated a greater occurrence of toxinotype A, but as I wrote in the conclusion Given the rich biodiversity of the Amazon biome and the diverse range of mammalian species that inhabit it, it is likely that C. perfringens is present in many of these animals. However, more research is needed to fully understand the prevalence and impact of this bacterium on wild mammal populations in the Amazon.
8). Discussion: This part should include the weaknesses of this study and work needs to be done in the future. New paragraph inserted in the discussion.

Thank you very much!
Sincerely,

Felipe Masiero Salvarani

Round 2

Reviewer 3 Report

Comments and Suggestions for Authors

Dear author thank you for submitting additional things and your reply. 

Please go through the following:

1) Please delete the PCR gel image from the manuscript.

2) All images of supplementary file (including PCR Gel image which is already there in supplementary file) put together in one word file, write down headlines as supplementary file, write the title of manuscript, authors and initial details, and set all image in that word file with proper number and captions (S.1, S.2, S.3) with describing the image including PCR gel image (please indicate/mark, numbering of gel image, Please see any standard published PCR gel image) and cite the number in main manuscript as and wherever is required.

(For any clarification or assistance, please go through any standard paper having supplementary file attached and how they cited in main manuscript!) As you are going to report for the first time, your manuscript needs to be more effective with all such data.

3) Please delete Anaerobic chamber image as you have already mention the manufecturer detail.

All the efforts is to make your manuscript more refine, crisp and effective. 

Comments on the Quality of English Language

minor

Author Response

Dear Reviewer 3,

We thank you for reviewing our article, suggesting important corrections to improve the article.

Below, point by point, I detail the corrections made following your suggestions.
1) Please delete the PCR gel image from the manuscript. Correction made.

2) All images of supplementary file (including PCR Gel image which is already there in supplementary file) put together in one word file, write down headlines as supplementary file, write the title of manuscript, authors and initial details, and set all image in that word file with proper number and captions (S.1, S.2, S.3) with describing the image including PCR gel image (please indicate/mark, numbering of gel image, Please see any standard published PCR gel image) and cite the number in main manuscript as and wherever is required. Correction made.

3) Please delete Anaerobic chamber image as you have already mentioned the manufacturer detail. Correction made.

Thank you very much for helping us make your manuscript more refined, crisp and effective. Your suggestions and guidance were fundamental to improving the manuscript, again, thank you very much.
